# *Piper nigrum* Extract: Dietary Supplement for Reducing Mammary Tumor Incidence and Chemotherapy-Induced Toxicity

**DOI:** 10.3390/foods12102053

**Published:** 2023-05-19

**Authors:** Nadeeya Mad-adam, Siribhon Madla, Narissara Lailerd, Poonsit Hiransai, Potchanapond Graidist

**Affiliations:** 1Department of Biomedical Sciences and Biomedical Engineering, Faculty of Medicine, Prince of Songkla University, Songkhla 90110, Thailand; 2Department of Physiology, Faculty of Medicine, Chiang Mai University, Chiang Mai 50200, Thailand; 3School of Allied Health Sciences, Walailak University, Nakhon Si Thammarat 80160, Thailand; 4Center of Excellence in Marijuana, Hemp, and Kratom, Walailak University, Nakhon Si Thammarat 80160, Thailand

**Keywords:** anticancer, cancer prevention, low piperine fractional *Piper nigrum* crude extract (PFPE), tumor incidence

## Abstract

A low piperine fractional *Piper nigrum* extract (PFPE) was prepared by mixing cold-pressed coconut oil and honey in distilled water, namely, PFPE-CH. In this study, PFPE-CH was orally administered as a dietary supplement to decrease the risk of tumor formation and reduce the side effects of chemotherapeutic drugs during breast cancer treatment. The toxicity study demonstrated no mortality or adverse effects after administrating PFPE-CH at 5000 mg/kg during a 14-day observation period. Additionally, PFPE-CH at 86 mg/kg BW/day did not cause any harm to the kidney or liver function of the rats for six months. In a cancer prevention study, treatment with PFPE-CH at 100 mg/kg BW for 101 days induced oxidative stress and increased the immune response by altering the levels of cancer-associated cytokines (IL-4, IL-6, and IFN-g), leading to a reduction in the tumor incidence of up to 71.4% without any adverse effects. In combination with doxorubicin, PFPE-CH did not disrupt the anticancer effects of the drug in rats with mammary tumors. Surprisingly, PFPE-CH reduced chemotherapy-induced toxicity by improving some hematological and biochemical parameters. Therefore, our results suggest that PFPE-CH is safe and effective in reducing breast tumor incidence and toxicity of chemotherapeutic drugs during cancer treatment in mammary tumor rats.

## 1. Introduction

Breast cancer is one of the most common cancers affecting women worldwide. Among females, breast cancer is the most diagnosed cancer, with an estimated 2.3 million new cases (11.7%), and it was the leading cause of death with an estimated 0.7 million deaths (6.9%) in 2020 [1]. The most frequent chemotherapy regimens for breast cancer are doxorubicin, paclitaxel, cyclophosphamide, methotrexate, and 5-fluorouracil [2]. Adjuvant chemotherapy and radiation therapy are the principal treatment options for breast cancer patients. However, the adverse effects, including immunosuppression, myelosuppression, gastrointestinal toxicity, post-treatment toxicity, tumor relapse, and cancer drug resistance, limit their use in a clinical setting [3,4]. The poor side effect profile of cytotoxic pharmaceuticals has considerably diminished the value of therapeutic drugs.

Chemoprevention is a growing area of anticancer research with an increasing number of preclinical and clinical studies of several cancer types [5]. Chemoprevention, a means of cancer control that entirely prevents the occurrence of the disease, slows down or is reversed with the use of nontoxic natural, dietary, or synthetic products, and has emerged as a promising and pragmatic medical approach to reducing the risk of cancer [6]. There are three types of cancer chemoprevention, depending on the preventive mechanism. Primary chemoprevention is implemented to block the development of premalignant lesions, whereas secondary chemoprevention focuses on suppressing the progression of these lesions to cancer. Tertiary chemoprevention aims to prevent the relapse or dissemination of primary cancer [7]. The basic properties of identified chemopreventive substances are anti-inflammatory and anti-mutagenic properties capable of inhibiting proliferation and inducing apoptosis, which are critical criteria for their anticancer activity [8]. An appreciable amount of natural substances consisting of phytochemicals and dietary substances have been found to attenuate breast cancer by affecting cell proliferation, cell differentiation, angiogenesis, apoptosis, and cellular transduction pathways [9]. Numerous dietary and botanical natural products, such as curcumin, resveratrol, tryptanthrin, kaempferol, gingerol, emodin, quercetin, genistein, and epigallocatechin gallate have been shown to suppress the early and late stages of carcinogenesis [10]. Based on their ability to regulate multiple survival pathways safely and with low toxicity, there is considerable scientific interest in investigating them for cancer prevention. Therefore, advances in the chemoprevention field over the past years have been very inspiring, and probable applications of several dietary supplements and natural substances for the prevention of breast cancer have been evaluated in clinical studies [11].

The most well-known species in the pepper family of Piperaceae is *Piper nigrum* L., which can be used for anti-apoptotic, anti-inflammatory, anti-mutagenic, anti-metastatic, anti-tumor, and hepatoprotective activities [12]. The chemical constituents of *P*. *nigrum* are aromatic essential oils, alkaloids, amides, propenylphenols, lignans, terpenes, flavones, and steroids [13]. Interestingly, the ethanolic crude extract of *P*. *nigrum* exhibited anticancer effects against colorectal carcinoma [14], breast cancer cell lines, and adenocarcinoma in a mouse model [15]. In addition to that, the dried fruits of *Piper longum* have also been reported to have chemopreventive efficacy by suppressing cell proliferation in DMBA-induced oral carcinogenesis [16]. Moreover, piperlongumine from the roots of long pepper, *P*. *longum*, has significant chemotherapeutic and chemopreventive potential against cancers such as multiple myeloma, melanoma, pancreatic cancer, colon cancer, oral squamous cell carcinoma, non-small cell lung cancer, gastric cancer, biliary cancer, and prostate cancer [17]. In chili peppers, the pungent alkaloid capsaicin has been reported as a chemopreventive, tumor-suppressing, radiosensitizing, and anticancer agent in various cancer models [17].

In our previous studies, low piperine fractional *P*. *nigrum* extract (PFPE) showed cytotoxicity against breast cancer MCF-7 cells with an IC_50_ value of 7.45 µg/mL and inhibited tumor growth in N-nitrosomethylurea (NMU)-induced mammary tumorigenesis in rats without liver and kidney toxicity [18]. The mechanisms of PFPE were found to upregulate p53 and downregulate estrogen receptors (ER), E-cadherin, matrix metalloproteinase-9 (MMP-9), MMP-2, c-Myc, and vascular endothelial growth factor (VEGF), both in vitro and in vivo studies [19]. Moreover, PFPE-treated groups of NMU-induced rats suppressed the tumor induction rate by 80% and 90% at 100 mg/kg and 200 mg/kg BW, respectively [18]. Additionally, PFPE has been reported to have anticancer immunity via promoting Th1 cells (anticancer) and suppressing Th2 and Treg cells (cancer promotion). PFPE can also suppress the levels of cytokines and chemokines linked to cancer development [20]. Furthermore, PFPE combined with doxorubicin enhanced and returned blood parameters to normal ranges in mammary tumor-bearing rats [21]. From a review of the literature and our previous reports, this plant extract has chemo-preventive properties, which are potent enough to decrease the risk of tumor formation and inhibit tumor growth in both in vitro and in vivo studies. Therefore, the functional formulation consisting of PFPE dissolved in cold-pressed coconut oil and honey was investigated in combination with the chemo-therapeutic drug (doxorubicin) in order to evaluate the chemo-preventive effects and safety information of PFPE (acute toxicity and long-term treatment).

## 2. Materials and Methods

### 2.1. Preparation of PFPE

The black peppercorn (*P*. *nigrum* L.) was collected from Songkhla Province, Thailand. The plant specimen (voucher specimen number SKP 146161401) was identified by Assistant Professor Supreeya Yuenyongsawad and deposited in the herbarium at the Southern Centre of Thai Traditional Medicine, Department of Pharmacognosy, and Pharmaceutical Botany, Prince of Songkla University, Thailand. PFPE was prepared according to the method described by Sriwiriyajan et al. [18]. Briefly, 70 g of the dry powder of black peppercorn was macerated in 200 mL of dichloromethane in a shaker incubator for 3 h. Then, the solution was filtrated using Whatman filter paper (No.1) (Whatman Bioscience, Ely, UK) and concentrated with a rotary evaporator (BUCHI, CH) at 45 °C for 3 h. Piperine in the dark brown oil residue of the extracts was crystallized by adding 100 mL of cold diethyl ether. The yellow crystals (piperine) were filtered out using Whatman filter paper (No.3). The brown residue was then concentrated in a vacuum at 45 °C for 3 h using a rotary evaporator. The concentrated extract was then kept in a desiccator until it was required for further use. Each PFPE dose was prepared by mixing 5% cold-pressed coconut oil and 20% honey in distilled water, namely, PFPE-CH.

### 2.2. Phytochemical Analysis and Identification of Bioactive Constituents Using Gas Chromatograph-Mass Spectrometer (GC-MS)

The chemical constituents of PFPE, including flavonoids, tannins, alkaloids, steroids, phenols, glycosides, lignans, and terpenoids, were analyzed using the Gas Chromatography Agilent 7890B combined with an Agilent 5977A triple quadrupole mass spectrometer (Agilent Technologies Inc., Santa Clara, CA, USA). The GC-MS detection was carried out as previously described [22]. All procedures were performed at the Scientific Equipment Center, Prince of Songkla University, Songkhla, Thailand. The PFPE components were identified by comparing their mass spectra with those in the GC-MS system software version wiley10 and NIST14, and the relative concentration of each compound in PFPE was further quantified based on the peak area integrated with the analysis program.

### 2.3. Animals

The female imprinting control region (ICR) mice (56 days old) and female Sprague Dawley rats (50 days old), which weighed 30–35 g and 150–180 g, respectively, were obtained from the National Laboratory Animal Center, Nakorn Pathom, Thailand. These animals were handled at the Southern Laboratory Animal Facility, Prince of Songkla University, Hat Yai, Songkhla, Thailand, at a room temperature of 25 ± 1 °C and with a 12 h light/dark cycle. Food and water were fed to animals ad libitum. Before beginning the trials, the mice and rats were given a 7-day acclimatization period to help them adjust to the laboratory environment. All experiments were conducted under the approval of the Institutional Animal Care and Use Committee (Ref. 32/2017), Prince of Songkla University, Thailand.

### 2.4. Experimental Design

#### 2.4.1. Acute Toxicity

The oral acute toxicity study of the PFPE-CH was carried out following Test No. 425 of the Organization for Economic Co-operation and Development (OECD) guideline [23]. The ICR mice were randomly divided into 3 groups with a maximum of 5 mice per group. Group 1 (control) received distilled water. Group 2 (vehicle) received a mixture of 5% cold-pressed coconut oil and 20% honey (CH) in distilled water. Group 3 (test group) received 5000 mg/kg body weight (mg/kg BW) of PFPE-CH. Each animal was then observed for symptoms of toxicity and mortality for the first 4 h after dosing and then daily for 14 days. The animals were monitored for 14 days to determine the long-term possible lethal outcomes.

#### 2.4.2. Chemoprevention Study

Rats were randomly separated into 6 groups of 10 animals each. Animals in each group were injected intraperitoneally with 150 mg/kg BW of NMU (Toronto Research Chemicals, Toronto, ON, Canada) to develop breast tumors at 50 and 80 days of age. Group 1 (control) did not receive any therapy. Group 2 (vehicle) received a mixture of 5% cold-pressed coconut oil and 20% honey in distilled water. Groups 3 to 5 (single preventive treatment) were orally treated with PFPE-CH at doses of 100, 200, and 250 mg/kg BW, respectively. Group 6 (double combination preventive treatment) was orally treated with 100 mg/kg BW of PFPE-CH mixed with 25 mg/kg BW of turmeric (Tu). After 14 days of the initial NMU application, animals in the treatment group were treated orally thrice weekly for 101 days. The rats were palpated twice weekly to detect mammary tumors.

#### 2.4.3. Chemotherapeutic Study

Rats were randomly separated into 8 groups of 10 animals each. Group 1 (control) received no therapy. Group 2 (vehicle) received 5% cold-pressed coconut oil and 20% honey in distilled water. Group 3 (positive control) received doxorubicin (Dox) (Adriamycin^®^, Pfizer Thailand LTD, Bangkok, Thailand) at a dose of 2 mg/kg BW. Groups 4 and 5 (single treatment) were given PFPE-CH orally at doses of 100 and 200 mg/kg BW, respectively. Groups 6 and 7 (double combination treatment) were given DOX at a dose of 2 mg/kg BW, followed by orally administered PFPE-CH at doses of 100 or 200 mg/kg BW, respectively. Group 8 (triple combination treatment) was given 100 mg/kg BW of PFPE-CH plus 25 mg/kg BW of Tu, followed by 2 mg/kg BW of DOX. Animals in each group were given 150 mg/kg BW of NMU to induce breast cancer at 50 and 80 days of age. Upon discovery of an initial tumor with a diameter of 2–3 mm, the indicated treatment conditions were administered 3 times a week for 30 days. Tumor volumes were measured as previously described [18]. At the end of the experiment, all rats were sacrificed, and wet tumors and organs were weighed. After blood was collected from the rats via heart puncture, hematological and biochemical analyses, oxidative stress markers, and cytokine studies were performed.

#### 2.4.4. Chronic Toxicity Study

Rats were randomly divided into 6 groups of 10 animals each. Group 1 (Control) received no treatment. Group 2 (vehicle) received 5% cold-pressed coconut oil and 20% honey in distilled water. Groups 3 to 5 (single treatment) were given PFPE-CH orally at doses of 43, 86, and 108 mg/kg BW. Group 6 (combination treatment) was administered 43 mg/kg BW of PFPE-CH plus 11 mg/kg BW of Tu. The treated group was orally given the PFPE-CH extract daily for 180 days (6 months). All rats were observed for apparent signs of toxicity during the experimental period. At the end of the experiment, blood was collected from the rats for hematologic and clinical chemistry study. The organs (heart, kidney, liver, lung, spleen, and stomach) were weighed; the organ weight/body weight ratio was calculated.

### 2.5. Histopathological Study

Mammary tissues were collected from each animal group and fixed in 10% buffered formalin. The tissues were then processed sequentially and washed with phosphate-buffered saline, dehydrated in an ascending series of ethanol, and cleared with acetone and xylene. Furthermore, tissues were embedded in a paraffin block and each block was cut into 5 µm. The specimens were cleaned and deparaffinized with xylene. The slides were sequentially rehydrated in ethanol and rinsed in running water before and after being dipped in hematoxylin dye. The sections were then stained with eosin solution. After being air-dried, they were mounted with DPX (mounting media) and covered with a glass slip; the slides underwent a histopathologic investigation.

### 2.6. Thiobarbituric Acid Reactive Substances (TBARS)

TBARS assay (R&D Systems, Minneapolis, MN, USA) was used to evaluate the lipid peroxidation in biological fluids, which considered the levels of oxidative stress within a biological sample. Briefly, 50 µL of MDA standard (0–64 mM) or sample was mixed with 50 mL of SDS lysis solution and incubated at room temperature for 5 min. After adding the thiobarbituric acid (TBA) reagent, the samples were incubated for 60 min at 95 °C. Next, the samples were centrifuged at 3000× *g* for 15 min and transferred supernatant to a new tube. Then, an equal volume of n-butanol was added, and the mixture was processed for another 1–2 min. The sample was centrifuged at 10,000× *g* for 5 min, and the butanol fraction was then measured at a wavelength of 532 nm using a spectrophotometer (Thermo Fisher Scientific, Inc., Watham, MA, USA). The MDA concentration of the sample was calculated as previously described [19].

### 2.7. Cytokine Detection Assay

At the end of the experiment, the blood samples were collected from each animal group. Serum concentrations of interleukin 4 (IL-4), IL-6, IL-10, and interferon γ (IFN-γ) were determined using Cytokine & Chemokine 22-Plex Rat ProcartaPlex^TM^ Panel (Thermo Fisher Scientific, Watham, MA, USA). The protocol was conducted according to the manufacturer’s instructions. Samples were analyzed using the Luminex 200 detection system (Thermo Fisher Scientific, Watham, MA, USA), and the data were analyzed using xPONENT MAGPIX software (Thermo Fisher Scientific, Watham, MA, USA).

### 2.8. Statistical Analysis

The results of all experiments were expressed as mean ± S.E.M., except for cytokine results, which were expressed as mean ± S.D. The statistical significance of the difference was estimated by one-way ANOVA. Differences were considered significant at *p*-values less than 0.05.

## 3. Results

### 3.1. GC-MS Analysis Revealed That 41 Compounds Contain PFPE

To quantify the phytochemical contents and chemical compound constituents, the PFPE was analyzed using the GC-MS technique. The composition of PFPE based on mass spectra and retention time indicated the presence of more than 40 phytochemical constituents in 4 groups, including alkaloids (52.822%), terpenes (27.125%), amides (19.650 %), and lignans (0.492%). Among 41 phytochemicals, more than 30 constituents contributed to antioxidant, antimicrobial, antifungal, and anticancer activities (Appendix A [24,25,26,27,28,29,30,31,32,33,34,35,36,37,38,39,40,41,42,43,44,45,46,47,48,49,50,51,52,53,54,55,56,57,58,59,60,61,62,63,64,65,66,67,68,69,70,71,72,73]). Due to the crystallization step in the PFPE extraction protocol, the percentage of piperine in the dichloromethane extract of *P*. *nigrum* crude extract remained at 19.59%, whereas the extracted solvents, diethyl ether, and dichloromethane were less than 4 femtograms.

### 3.2. PFPE-CH Does Not Induce Acute Toxicity in ICR Mice

The OECD Test Guideline’s Up and Down Procedure was used to assess the acute toxicity test at an oral limit dose of 5000 mg/kg BW of the PFPE-CH formulation. After the first 4 h period, the ICR mice did not demonstrate acute toxic signs. Thus, daily monitoring of the long-term results was performed. Throughout the 14-day observation period, the PFPE-CH-treated ICR mice displayed normal behavior with no signs of toxicity. Furthermore, none of the groups had a lower body weight or food intake than the control group (Figure 1A,B). However, the vehicle and PFPE-CH groups had less effect on water consumption than the control group (Figure 1C). These findings suggest that PFPE-CH is safe for acute toxicity and that the oral LD_50_ in mice is greater than 5000 mg/kg BW in ICR mice.

### 3.3. PFPE-CH Reduces Mammary Tumor Rat Incidence

To determine the cancer-preventive effect of PFPE-CH, rats were intraperitoneally injected with 2 doses of NMU and then orally treated with PFPE-CH for 101 days. In this study, at least 10% of the rats in both the control and vehicle groups developed their first detectable breast tumor within 37 days of the study period. Rats treated with PFPE-CH at a dose of 100 mg/kg BW developed the first tumor of 10% on day 44 and accumulated to 20% on the sacrificed date, whereas only a single rat treated with PFPE at doses of 200 and 250 mg/kg BW was found to have a tumor on day 79 and 44, respectively. Furthermore, the combination of 100 mg/kg BW PFPE-CH with 25 mg/kg BW turmeric resulted in a tumor incidence of 40% in treated animals, with the first tumor appearing on day 58. Moreover, the single PFPE-CH treatment or the combination with turmeric showed no significant change in body weight, food, and water intake from day 0 to day 79 compared to the control group. However, PFPE-CH at doses of 250 mg/kg BW showed a significant difference in reduced food intake compared to the control group on days 80, 87, and 94 of the treatment period (Figure 2). Moreover, the organ weight ratio in PFPE-CH treated groups did not significantly change compared to the control and vehicle groups (Table 1).

In the leukocyte differentiation, PFPE at a dose of 250 mg/kg BW significantly decreased lymphocyte proportion while the neutrophil proportion significantly increased (Table 1). Additionally, the clinical blood chemistry showed that PFPE at a dose of 250 mg/kg BW significantly increased the level of blood urea nitrogen (BUN) and creatinine while decreasing the level of alkaline phosphatase. Moreover, PFPE-CH at a dose of 200 mg/kg BW affected the reduction in SGOT activity. Therefore, PFPE-CH at a dose of 100 mg/kg BW is the highest dose with no observed adverse effects on behavioral health.

### 3.4. PFPE-CH Did Not Disrupt the Anticancer Effects of Doxorubicin on Mammary Tumor Rats

In a cancer therapeutic experiment, the anticancer effect of PFPE-CH alone and in combination with chemotherapeutic medication (doxorubicin) was assessed by evaluating tumor growth. The tumor volume of the control and vehicle groups increased in a time-dependent manner from day 0 to day 24 (Figure 3A). However, the tumor volume in the vehicle group significantly decreased compared to the control group. Only doxorubicin effectively decreased tumor development at all time points compared to the control group. In the only PFPE-CH therapy, both the doses of 100 and 200 mg/kg BW suppressed tumor development from day 0 to day 18, and significant enlargement was observed on days 20 and 22, respectively. However, the tumor volume in the PFPE-CH-treated group was significantly lower than the vehicle group, when compared at the same time points from day 16 to day 24. On the other hand, the same phenomenon was not observed in the combination treatment. At the last 5 time points (from day 16 to day 24), the tumor volume of each treatment group was normalized with the mean tumor volume of the control group and expressed as the percentage of tumor suppression (Figure 3B). PFPE-CH at a dose of 100 mg/kg BW showed tumor suppression rates comparable to those of doxorubicin from day 0 to day 18, and thereafter gradually declined from day 20 to day 24. Doxorubicin therapy and the combination of doxorubicin and PFPE-CH did not significantly differ in terms of tumor suppression rates. However, the percentage of tumor reduction rate was decreased in the triple combination treatment. Interestingly, the PFPE-CH at doses 100 and 200 mg/kg BW did not significantly change both the body weight and organ weight/body weight ratio compared to the control and vehicle groups. Meanwhile, doxorubicin significantly increased the liver weight ratio and reduced the spleen weight ratio, as shown in Table 2. However, the size of the spleen returned to normal when treated with PFPE-CH. In contrast, the combination of doxorubicin with 100 mg/kg BW PFPE and 25 mg/kg BW turmeric significantly decreased the spleen weight ratio.

H&E staining of tumor histology, as shown in Figure 3C–K, revealed that the terminal duct lobule (TDL) and fatty tissue were present in the typical histopathological characteristics of normal breast tissue (Figure 3C). A strong basement membrane encircled the ductal epithelial and myoepithelial cells that made up the terminal duct. In contrast, the tumor histological characteristics of the control (Figure 3D) and vehicle (Figure 3E) groups showed tightly packed cells with high-grade ductal carcinoma and invasive ductal carcinoma, with 67–80% of grade III and 20–33% of grade I-II (Appendix A [74]). Rats that were given doxorubicin treatment demonstrated a reduction in ductal carcinoma to 17% in grade III and 17% in grade I (Figure 3F). In the PFPE-CH group that received a dose of 100 mg/kg BW, the ductal carcinoma initially showed 100% grade III (Figure 3G) manifestation, which decreased to 50% when the dose was 200 mg/kg BW (Figure 3H). The combination of doxorubicin with PFPE-CH at doses of 100 (Figure 3I) and 200 mg/kg BW (Figure 3J) reduced grade III histology to 33% and 17%, respectively. However, the triple combination revealed 100% of grade III carcinoma (Figure 3K). These findings suggest that PFPE-CH at 200 mg/kg BW is the lowest dose that can reduce breast tumor histology and is more effective when combined with drug therapy than turmeric.

The results from blood parameter analysis demonstrated that treatment with doxorubicin alone and the triple combination treatment affected white blood cells, hemoglobin, hematocrit, total protein, and albumin in rats. Notably, no significant differences in the hematologic and clinical blood chemistry values were observed in the rats treated with PFPE-CH at doses of 100 mg/kg BW compared to the control group. Interestingly, rats that received a combination treatment of doxorubicin and PFPE-CH showed normal levels of white blood cells. However, hemoglobin and hematocrit levels in the combination treatment of doxorubicin with PFPE-CH groups increased compared to those treated with doxorubicin alone (Table 2). These results suggest that PFPE-CH at a dose of 100 mg/kg BW is the highest dose that can inhibit cancer cell growth without any adverse effects on hematological and blood biochemical values. In addition, the combination of PFPE-CH with doxorubicin increased white blood cells, red blood cells, hemoglobin, hematocrit, platelets, total protein, and alkaline phosphatase compared to doxorubicin treatment alone.

### 3.5. PFPE-CH Increased the Stress Condition, Indicated by the Elevation of TBARS Level

At the end of the prevention study period, blood was drawn from the heart on the sacrifice day. Serum samples from each group were pooled and oxidative stress conditions were determined. The results show that in the presence of PFPE-CH at doses of 200 and 250 mg/kg BW and PFPE-CH in combination with turmeric, the TBARS level significantly increased compared to the control and vehicle groups. The increase in TBARS was positively correlated with the tumor observation time (Figure 4A). The results indicate that PFPE-CH at doses of 200 and 250 mg/kg BW, as well as PFPE-CH combined with turmeric, showed cancer prevention effects through the generation of oxidative stress to induce cancer stress, which can inhibit tumor growth.

On the 30th day after treatment, the serum was collected in order to determine the TBARS level, as shown in Figure 4B. The TBARS level of doxorubicin increased compared to that of the control and vehicle groups. In contrast, the TBARS level decreased at doses of 100 and 200 mg/kg BW of PFPE-CH treatment. However, the TBARS level of the PFPE-CH combination with doxorubicin was higher compared to that of the vehicle and control groups. On the contrary, TBARS levels decreased in the triple combination group compared to that of the control and double combination groups. These findings suggest that PFPE-CH at a dose of 100 mg/kg BW is the lowest dose that enhances the effect of doxorubicin via increased oxidative stress in cancer cells, resulting in DNA damage and cell death without any adverse effects. However, the triple combination seems to decrease the effectiveness of doxorubicin in eliminating cancer cells.

### 3.6. PFPE-CH Altered the Levels of IL-4, IL-6, IL-10, and IFN-γ

In the prevention study, rat serum determined the levels of IL-4, IL-6, IL-10, and IFN-γ compared to those in the control and vehicle groups. As shown in Figure 5, the levels of IL-4, IL-6, and IFN-γ decreased with PFPE-CH therapy at all treatment dosages. Curiously, the combination of PFPE-CH and turmeric significantly boosted the levels of IL-10 but only marginally increased the levels of IL-4 and IFN-γ, whereas the IL-6 level remained unchanged. These results indicate that treatment with PFPE-CH alone had a cancer prevention effect by reducing the levels of IL-4, IL-6, and IFN-γ, as well as IL-6-associated TBARS levels.

For the cancer treatment study, IL-4 levels in the double and triple combination treatments significantly decreased compared to those in the control and vehicle groups (Figure 6A). The level of IL-6 significantly increased in PFPE-CH at a dose of 100 mg/kg BW. However, IL-6 levels decreased when combining PFPE-CH and doxorubicin. Meanwhile, IL-6 levels increased in the triple combination group compared to the control and vehicle groups (Figure 6B). Treatment with PFPE-CH at a dose of 100 mg/kg BW significantly increased the levels of IL-10 compared to the control and vehicle groups. However, the level of IL-10 decreased in the double combination treatment but not in the triple combination treatment (Figure 6C). The levels of IFN-γ in the vehicle and doxorubicin treatment were higher compared to those in the vehicle group. Meanwhile, the IFN-γ levels in the presence of PFPE-CH were significantly lower compared to those in the control and vehicle groups. However, IFN-γ level increased in the double and triple combinations compared to a single treatment with PFPE-CH (Figure 6D). These results indicate that a single treatment with PFPE-CH at a dose of 200 mg/kg BW can reduce breast tumor volume and tumor histology by decreasing IL-4 and IL-6 levels and is more effective when combined with doxorubicin.

### 3.7. PFPE-CH Does Not Cause Chronic Toxicity in Sprague Dawley Rat

A cancer-preventive study of PFPE-CH showed that giving PFPE-CH at doses of 100, 200, and 250 mg/kg BW three times a week to rats could prevent cancer development. Therefore, these doses were fed to rats daily in order to test the chronic toxicity at doses of 43, 86, and 108 mg/kg BW/day, respectively. The experiment lasted for more than 6 months, and throughout the entire trial period, no death occurred among the tested rats. The body weight and organ weight of the PFPE-CH group did not demonstrate any significant differences compared to the control group. However, rats treated with 108 mg/kg BW/day of PFPE-CH showed a reduced level of serum SGOT compared to the control and vehicle groups. Moreover, rats treated with the combination treatment exhibited reductions in serum creatinine levels compared to the control and drug groups (Table 3). Therefore, the suitable doses of PFPE-CH for long-term oral application were daily doses of 43 and 86 mg/kg BW/day or 100 and 200 mg/kg BW three times a week.

## 4. Discussion

Many dietary natural products can affect the development and progression of breast cancer. *P*. *nigrum*, the most well-known species of pepper, has been reported to have anti-inflammatory, anti-mutagenic, and anticancer activities [12]. Piperine is the major bioactive compound identified in *P*. *nigrum*. However, the PFPE used in our study was piperine removed from crude extracts by recrystallization with cold diethyl ether. The results from GC-MS analysis (Appendix A) revealed that compounds containing PFPE showed antiproliferative activity against several human cancer cells, including caryophyllene, piperanine, piperolein B, pipersintenamide, beta-bisabolene, piperettine, and (−)-kusunokinin [28,36,60,75]. Therefore, these compounds show the anticancer activities of PFPE.

In this study, the potential use of PFPE-CH as an oral dietary supplement was investigated in order to reduce tumor incidence and support the body during cancer treatment. PFPE is poorly dissolved in water leading to limited oral bioavailability. Thus, PFPE was dissolved in a mixture of 5% cold-pressed coconut oil with 20% honey in distilled water to form the PFPE-CH formulation. Cold-pressed coconut oil and honey were chosen because they are natural health products, abundantly available, low cost, and biocompatible. Coconut oil has been reported to inhibit carcinogenic agents in the colon and mammary tumor animal models [76,77]. Consuming coconut oil during chemotherapy improves the functional status and quality of life of breast cancer patients [78]. Honey contains vitamins, phenolic acids, flavonoids, proteins, carbohydrates, amino acids, and royal jelly aliphatic acids [79,80], which have health-promoting properties, including cough reduction and antioxidant, anti-inflammatory, antibacterial, antiviral, antiparasitic, antimutagenic, and antitumor properties [81,82]. Moreover, adding honey to the formulation reduced the spiciness of PFPE.

In this study, PFPE-CH was tested for its efficacy in tumor-growth prevention and treating breast cancer in rat models at doses of 100, 200, and 250 mg/kg BW. Curcumin, the primary bioactive substance in turmeric, is an active compound from *Curcuma longa* L. and “generally recognized as safe” as categorized by the FDA, and it is widely used in complementary therapy for cancer patients in combination with chemotherapeutic drugs [83]. Therefore, the combination of 100 mg/kg BW of PFPE-CH and 25 mg/kg BW of turmeric was used. The protective effect of PFPE-CH was approximately 100, 200, and 250 mg/kg BW and was able to inhibit breast cancer occurrence at 71%, 80%, and 80%, respectively. Meanwhile, the combination of PFPE with turmeric showed an inhibition rate of 43%.

We further evaluated the effect of PFPE-CH on the relationship between tumor formation, reactive oxygen species (ROS), and cancer-related cytokines. Cytokines regulate key immune players in cancer development and progression and are potential targets for treatment [84]. In this study, PFPE-CH effectively induced oxidative stress and decreased the levels of IL-4 and IL-6. These results have positive correlations with the observed tumor time. Moreover, the PFPE-CH group had only a 20% tumor incidence and could slow the onset of cancer compared to the control and vehicle groups. The first detectable tumors in the combination therapy were found later than with PFPE-CH at 100 mg/kg BW alone. However, the number of mammary tumor rats gradually increased to 40% after the first tumor was detected. These results might be consistent with the highest levels of IL-4 and IL-10 that were found under this condition. Breast cancer progression is facilitated by IL4, a pleiotropic cytokine secreted by fibroblasts, Th2, adipose-derived stem cells (ADSCs), as well as breast cancer cells. IL-4 promotes metastatic spreading by activating the MAPK pathway [85]. Moreover, increased IL-10 secretion is associated with aggressive breast cancer via the transformation of M1 macrophage to an M2 angiogenic phenotype [86]. For this reason, cancer cells could evade immunosurveillance and form tumors that were detected during the experiment. However, curcumin is a potent immunomodulatory agent. In breast cancer, curcumin can increase M1 and decrease M2 macrophages, resulting in a decrease in STAT3, IL-10, and arginase I gene expression and secretion in mice with metastatic breast cancer [87]. In our study, turmeric combination with PFPE-CH contradicts the results of a previous study that reported that piperine enhances the absorption of curcumin into the bloodstream and can improve the activity of curcumin [88]. We found that the combination of curcumin and PFPE was less effective in preventing cancer than PFPE-CH alone. This phenomenon could be due to PFPE having a low level of piperine, which might cause low absorption of curcumin, resulting in less effectiveness of this combination in this study. Herein, we concluded that PFPE-CH alone had a secondary chemopreventive effect by decreasing tumor formation, leading to a reduction in tumor incidence.

In addition, PFPE-CH at a dose of 250 mg/kg BW significantly decreased the lymphocyte proportion, whereas the neutrophil significantly increased in the prevention experiment. Lymphocytes can destroy tumor cells without prior sensitization. However, many tumors downregulate the expression of class 1 major histocompatibility complex (MHC) molecules, causing reduced recognition of T lymphocytes and leading to a possible decrease in the lymphocyte count [89]. In the liver function test, PFPE-CH at a dose of 200 mg/kg BW reduced SGOT activity. On the other hand, PFPE-CH at a dose of 250 mg/kg BW significantly decreased the alkaline phosphatase compared to control rats. In fact, the low activity of SGOT and alkaline phosphatase was not a sign of progressive liver damage [90,91].

In part of the chemotherapeutic activity, doxorubicin treatment exhibited increased liver weight due to its side effects such as hematopoietic suppression and hepatotoxicity, and the most serious side effect is life-threatening cardiomyopathy [92]. In addition, doxorubicin also decreased the white blood cells, hemoglobin, hematocrit, total protein, and albumin. This drug causes bone marrow suppression, resulting in anemia, neutropenia, and thrombocytopenia [4]. Our results showed that doxorubicin treatment increased the INF-γ levels. In general, doxorubicin increases INF-γ in tumors by accumulating IL-12 and is a signal transducer and activator of transcription 1 (STAT1)-dependent antitumor efficacy [93]. Moreover, oxidative stress was increased after doxorubicin treatment, resulting in DNA damage and cancer cell death [94,95]. These results are consistent with our results that showed a decrease in tumor volume and the highest tumor suppression rate by reducing IL-4 and IL-6 levels. However, the vehicle treatment significantly decreased the tumor volume compared to the control group. This phenomenon could be due to the effects of honey and cold-pressed coconut oil, as mentioned above. Meanwhile, PFPE-CH alone significantly reduced tumor size compared to the vehicle group. Therefore, the reduction in tumor size was due to the effect of PFPE-CH, which contained various phytochemical compounds, as mentioned in Appendix A. Moreover, PFPE-CH has been reported to have anticancer activities in various pathways, both in vitro and in vivo [18,19,20,21]

The current study showed that PFPE-CH at a dose of 100 mg/kg BW had an inhibitory effect on tumor growth. However, the tumor volume increased after 18 days of treatment. At this point, we propose that it might be due to the components in PFPE that contain various types of compounds. Caryophyllene and piperine are the bioactive compounds found in PFPE with approximately 13% and 20%, respectively. Caryophyllene has several biological activities, including anti-carcinogenic, anti-inflammatory, and immune-modulatory. The remaining piperine in PFPE-CH also has therapeutic potential against many cancer types. Nevertheless, the pharmacological activities of caryophyllene and piperine are limited due to their low water solubility, high sensitivity to oxidation when exposed to light or oxygen, fast metabolism, and systemic elimination [96,97,98]. These properties could allow tumors to evade attacks from the immune system.

The inhibition of immune cells and soluble cytokines promotes tumor progression. For example, the upregulation of IL-6 increases tumor invasion and epithelial-mesenchymal transition in human breast cancer cells [99,100]. Moreover, cancer-associated fibroblasts (CAFs) from breast tissue can produce IL-10 that disrupts the cytokine balance to stimulate tumor growth by initiating angiogenesis [101]. In our current study, IL-6 and IL-10 levels were increased when treated with PFPE-CH at a dose of 100 mg/kg BW. This result was consistent with the tumor suppression rate at the last three time points (on days 20, 22, and 24). Doxorubicin and PFPE-CH have different tumor-suppressive mechanisms. The side effect of doxorubicin was reduced by PFPE-CH in the combination treatment. IL-4 and IL-10 levels decreased more in the combination treatment than doxorubicin or PFPE alone, which correlated with reduced tumor progression (Appendix A). Interestingly, the combination of doxorubicin with PFPE-CH slightly improved some blood parameters (white blood cells, red blood cells, hemoglobin, hematocrits, platelets, total protein, SGOT, and alkaline phosphatase).

Prior to exploring the possible uses of PFPE-CH as preventive and co-treatment agents, acute toxicity was determined according to the standard guidelines. A single oral dose of 5000 mg/kg PFPE-CH caused no death, no significant abnormal change in behavior, and no signs of toxicity signs in mice throughout the 14-day observation period. Therefore, PFPE-CH could be considered safe for acute toxicity, and its oral LD_50_ was greater than 5000 mg/kg in mice. Moreover, a chronic study reported that daily doses of 43, 86, and 108 mg/kg BW/day or 100, 200, and 250 mg/kg BW three times a week caused no deaths during the 6 months of the experimental period. PFPE-CH at a dose of 108 mg/kg BW/day was found to affect SGOT and alkaline phosphatase. Meanwhile, the combination of PFPE-CH and turmeric significantly decreased blood creatinine. A low level of creatinine is associated with liver disease and impaired renal function [102]. Therefore, the use of PFPE-CH in combination with turmeric is a concern. These findings suggest that PFPE at a dose of 200 mg/kg BW or 86 mg/kg BW/day was the highest dose that had no adverse effects on behavioral health for 6 months.

## 5. Conclusions

The PFPE-CH formulation exhibited secondary chemoprevention type and anticancer effects on NMU-induced rats. It could be considered safe for acute toxicity at doses of up to 5000 mg/kg and there was no evidence of any harm to rats at a dose of 86 mg/kg/day for 6 months. Its effects indicate the potential use of PFPE-CH as an orally administered dietary supplement to decrease the risk of breast tumor formation and support the body during cancer treatment processes. The recommended dose of PFPE administered alone was not more than 43 mg/kg/day or 100 mg/kg BW (3 times/week), which was the no-observed-adverse-effect level (NOAEL) for cancer prevention and anticancer treatment. In combination with PFPE and a chemotherapeutic drug (doxorubicin), the dose recommendation was not more than 86 mg/kg/day or 200 mg/kg BW (3 times/week). However, the effect of PFPE 200 mg/kg BW with doxorubicin on liver function is a concern. In addition, the triple combination of PFPE, curcumin, and doxorubicin is not recommended for use in cancer treatment.

## Figures and Tables

**Figure 1 foods-12-02053-f001:**
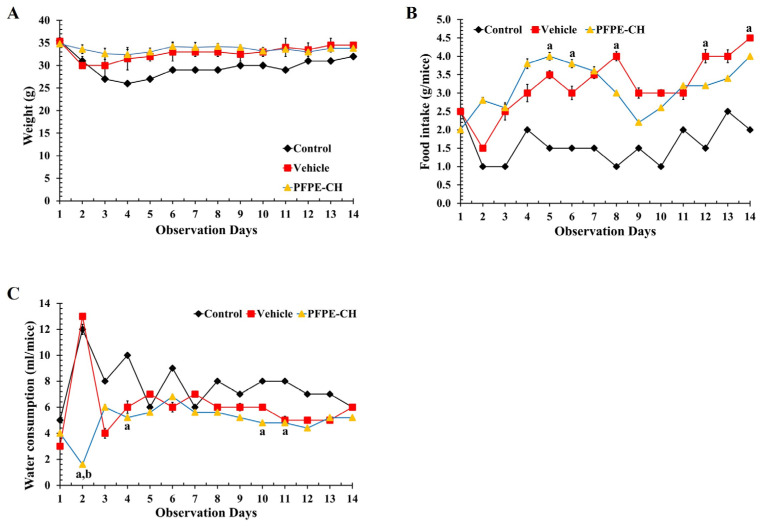
The effect of oral administration of PFPE-CH on the behavior health of mice for 2 weeks. (**A**) Body weight, (**B**) food intake, and (**C**) water consumption of mice that received PFPE-CH at an oral limit dose of 5000 mg/kg BW were recorded during the observation period. Values represent the mean ± SEM; ^a^
*p* < 0.05 was significantly different compared to the control group; ^b^
*p* < 0.05 was significantly different compared to the vehicle group using one-way ANOVA analysis.

**Figure 2 foods-12-02053-f002:**
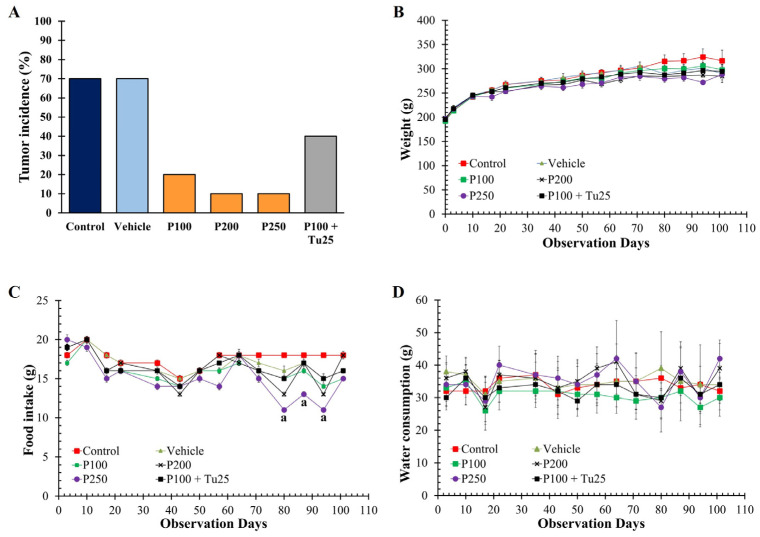
Effects of oral administration of PFPE-CH on a chemopreventive study in NMU-induced mammary tumor rats for 101 days. (**A**) Tumor incidence of rats treated with PFPE-CH, (**B**) body weight, (**C**) food intake, and (**D**) water consumption was recorded during a cancer prevention study in rats. ^a^
*p* < 0.05 was significantly different compared to the control group using one-way ANOVA analysis.

**Figure 3 foods-12-02053-f003:**
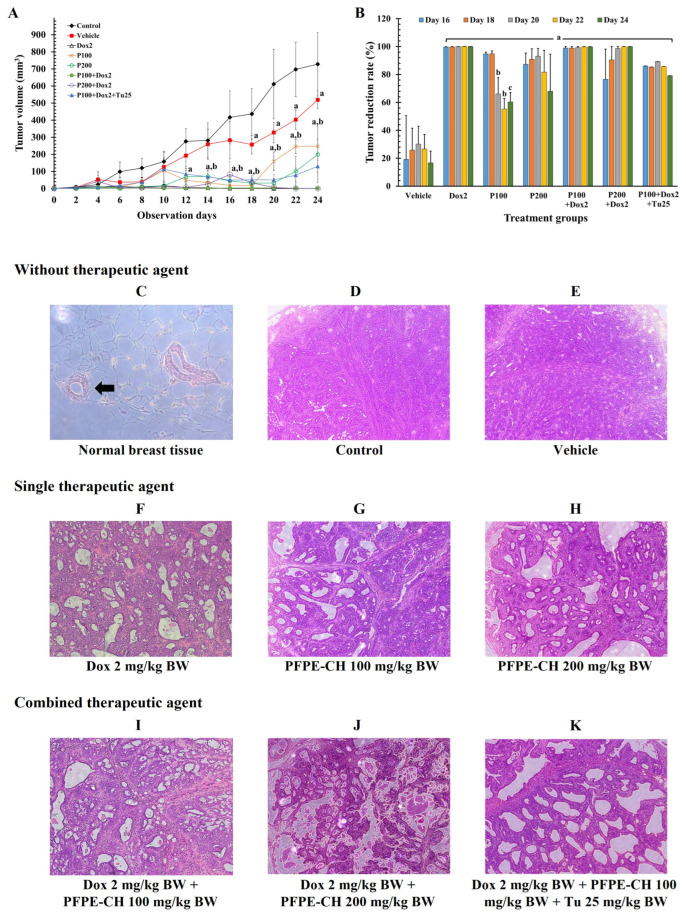
Effects of PFPE-CH and their combination treatment on the suppression of tumor growth and tumorigenesis rats. (**A**) The tumor volume of cancer treatment models during the observation period for 24 days. (**B**) The percentage of tumor reduction rate on days 16, 18, 20, 22, and 24 of treatments when normalized with the control group. (**C**–**K**) Paraffin-embedded sections were stained with hematoxylin and eosin. (**C**) Normal tissue was composed of the terminal duct-lobule unit and fatty tissue (arrow). The tumor tissue of (**D**) control and (**E**) vehicle groups represented densely packed cancer cells. (**F**–**K**) tumor tissue showed widespread necrosis in the tumor tissue. Dox: doxorubicin; Tu: Turmeric. Original magnification of (**A**) was 40×; original magnification of (**B**–**I**) was 10×. Dox2: Doxorubicin 2 mg/kg BW; P100: PFPE-CH 100 mg/kg BW; P200: PFPE-CH 200 mg/kg BW; P100 + Dox2: PFPE-CH 100 + Doxorubicin 2 mg/kg BW; P200 + Dox2: PFPE-CH 200 + Doxorubicin 2 mg/kg BW; P100 + Dox2 + Tu25: PFPE-CH 100 + Doxorubicin 2 + Turmeric 25 mg/kg BW. Data represent the mean ± SEM. ^a^
*p* < 0.05 was significantly different compared to the vehicle group; ^b^
*p* < 0.05 was significantly different compared to all groups; ^c^
*p* < 0.05 was significantly different compared to the Dox2, P100 + Dox2, and P200 + Dox2 groups of each time point using one-way ANOVA analysis.

**Figure 4 foods-12-02053-f004:**
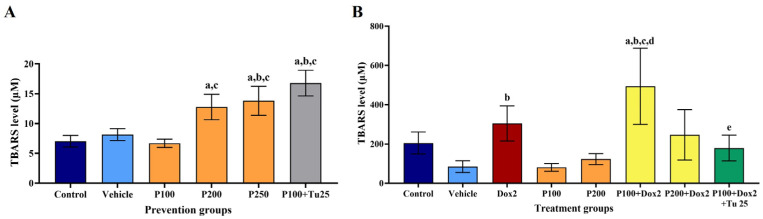
Effects of oral administration of PFPE-CH on oxidative stress conditions during (**A**) chemopreventive study and (**B**) chemotherapeutic study in rats. The TBARS levels, which indicated lipid peroxidation in blood samples, were investigated at the end of the experiment. P100: PFPE-CH 100 mg/kg BW; P200: PFPE-CH 200 mg/kg BW; P250: PFPE-CH 250 mg/kg BW; P100 + Tu25: PFPE-CH 100 + Turmeric 25 mg/kg BW; Dox2: Doxorubicin 2 mg/kg BW; P100 + Dox2: PFPE-CH 100 + Doxorubicin 2 mg/kg BW; P200 + Dox2: PFPE-CH 200 + Doxorubicin 2 mg/kg BW; P100 + Dox2 + Tu25: PFPE-CH 100 + Doxorubicin 2 + Turmeric 25 mg/kg BW. Data are shown as mean ± SEM. The a, b, c, d, and e were represented as *p* < 0.05, which were significantly different compared to the control, vehicle, P100, P200, and P100 + Dox2 groups, respectively, using one-way ANOVA.

**Figure 5 foods-12-02053-f005:**
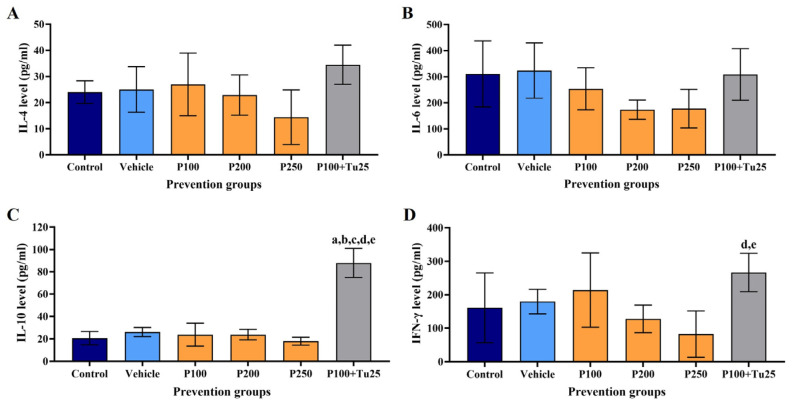
Effects of PFPE-CH on serum cytokines production in cancer prevention. (**A**) IL-4, (**B**) IL-6, (**C**) IL-10, and (**D**) IFN-γ levels of PFPE-CH-treated group in the cancer prevention study. The level of cytokines was evaluated in each group of NMU-induced mammary rats on the day of sacrifice. P100: PFPE-CH 100 mg/kg BW; P200: PFPE-CH 200 mg/kg BW; P250: PFPE-CH 250 mg/kg BW; P100 + Tu25: PFPE-CH 100 + Turmeric 25 mg/kg BW. Data are shown as mean ± SD. The a, b, c, d, and e were represented as *p* < 0.05, which were significantly different compared to the control, vehicle, P100, P200, and P250 groups, respectively, using one-way ANOVA.

**Figure 6 foods-12-02053-f006:**
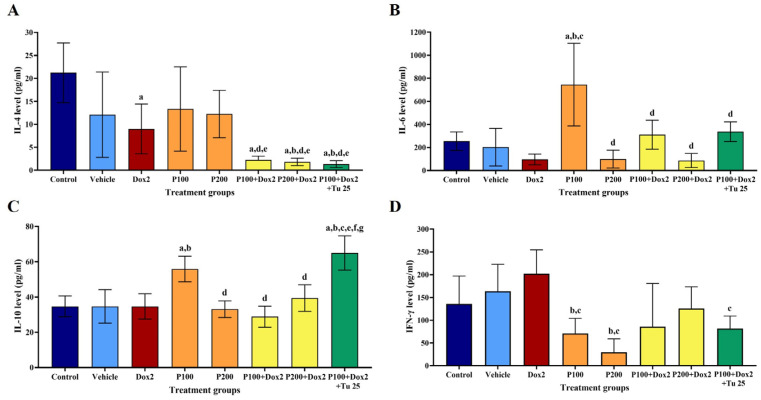
Effects of PFPE-CH on serum cytokines production in the cancer therapeutic study. (**A**) IL-4, (**B**) IL-6, (**C**) IL-10, and (**D**) IFN-γ levels of PFPE-CH-treated group in cancer therapeutic study. The level of cytokines was evaluated in each group of NMU-induced mammary rats on the day of sacrifice. Dox2: Doxorubicin 2 mg/kg BW; P100: PFPE-CH 100 mg/kg BW; P200: PFPE-CH 200 mg/kg BW; P100 + Dox2: PFPE-CH 100 + Doxorubicin 2 mg/kg BW; P200 + Dox2: PFPE-CH 200 + Doxorubicin 2 mg/kg BW; P100 + Dox2 + Tu25: PFPE-CH 100 + Doxorubicin 2 + Turmeric 25 mg/kg BW. Data are shown as mean ± SEM. The a, b, c, d, e, f, and g were represented as *p* < 0.05, which were significantly different compared to the control, vehicle, Dox2, P100, P200, P100 + Dox2, P200 + Dox2, and P100 + Dox2 + Tu25 groups, respectively, using one-way ANOVA.

**Table 1 foods-12-02053-t001:** Body weight, the organ weight to body weight ratio, and hematologic and clinical chemistry values of the tumor prevention study on the induced mammary tumorigenesis in rats.

Parameters	Control	Vehicle	PFPE-CH 100	PFPE-CH 200	PFPE-CH 250	PFPE-CH 100+ Tu 25
**Body weight** (**g**)	307.90 ± 8.34	305.40 ± 8.37	302.40 ± 6.77	297.00 ± 6.50	289.86 ± 10.68	302.60 ± 3.70
**Organ weight/body weight ratio**						
Heart	0.29 ± 0.01	0.30 ± 0.01	0.28 ± 0.00	0.31 ± 0.01	0.33 ± 0.01	0.29 ± 0.01
Kidney	0.63 ± 0.02	0.62 ± 0.02	0.61 ± 0.02	0.64 ± 0.01	0.64 ± 0.02	0.67 ± 0.02
Liver	3.45 ± 0.03	3.58 ± 0.19	3.33 ± 0.11	3.73 ± 0.12	4.08 ± 0.10	3.52 ± 0.12
Lung	0.43 ± 0.01	0.42 ± 0.01	0.40 ± 0.01	0.45 ± 0.01	0.46 ± 0.02	0.41 ± 0.01
Spleen	0.18 ± 0.01	0.19 ± 0.01	0.18 ± 0.01	0.19 ± 0.01	0.18 ± 0.01	0.19 ± 0.01
Stomach	0.42 ± 0.02	0.47 ± 0.03	0.45 ± 0.02	0.51 ± 0.02	0.51 ± 0.01	0.46 ± 0.02
**Hematologic values**						
White blood cells (×10^3^/µL)	3.38 ± 0.22	3.51 ± 0.16	3.54 ± 0.27	3.33 ± 0.25	3.01 ± 0.35	3.81 ± 0.23
Neutrophil (%)	30.80 ± 1.18	37.80 ± 3.65	31.0 ± 2.45	29.0 ± 3.21	46.42 ± 4.09 ^a^	32.60 ± 3.23
Lymphocyte (%)	67.80 ± 1.24	61.60 ± 3.83	69.60 ± 3.11	67.14 ± 5.56	53.57 ± 4.09 ^a^	67.20 ± 3.27
Monocyte (%)	0.90 ± 0.50	0.20 ± 0.13	0.00 ± 0.00	0.00 ± 0.00	0.00 ± 0.00	0.00 ± 0.00
Eosinophil (%)	0.50 ± 0.27	0.40 ± 0.27	0.00 ± 0.00	0.57 ± 0.57	0.00 ± 0.00	0.20 ± 0.20
Basophil (%)	0.00 ± 0.00	0.00 ± 0.00	0.00 ± 0.00	0.00 ± 0.00	0.00 ± 0.00	0.00 ± 0.00
Red blood cell (×10^6^/µL)	7.77 ± 0.27	7.89 ± 0.22	7.35 ± 0.23	7.44 ± 0.16	6.91 ± 0.41 ^a,b^	7.58 ± 0.17
MCV (fL)	54.40 ± 0.56	52.20 ± 0.47 ^a^	53.30 ± 0.45	52.71 ± 0.61 ^a^	53.86 ± 0.77	53.10 ± 0.43
MCH (pg)	19.11 ± 0.35	18.90 ± 0.23	19.30 ± 0.21	18.67 ± 0.21	20.86 ± 0.55	19.37 ± 0.46
MCHC (g/dL)	34.10 ± 1.73	36.60 ± 0.64	36.50 ± 0.37	33.57 ± 1.97	38.43 ± 1.13	33.20 ± 2.22
Hemoglobin (g/dL)	15.09 ± 0.60	14.93 ± 0.36	14.19 ± 0.45	14.30 ± 0.37	14.23 ± 0.67	14.79 ± 0.13
Hematocrit (%)	41.70 ± 1.73	40.70 ± 1.16	38.50 ± 1.20	38.86 ± 0.91	36.86 ± 1.97 ^a^	39.60 ± 0.81
Platelet (×10^3^/µL)	748.60 ± 33.60	754.70 ± 26.39	750.20 ± 25.79	783.29 ± 46.05	748.43 ± 25.53	795.40 ± 19.84
**Clinical blood chemistry values**						
BUN (mg/dL)	26.43 ± 1.45	28.03 ± 2.05	31.53 ± 1.43	31.71 ± 2.09 ^a^	34.61 ± 2.29 ^a,b^	31.89 ± 2.67
Creatinine (mg/dL)	0.66 ± 0.04	0.77 ± 0.03	0.85 ± 0.04	0.66 ± 0.05	0.83 ± 0.05 ^a^	0.73 ± 0.07
Total protein (g/dL)	6.36 ± 0.16	6.58 ± 0.19	6.56 ± 0.16	6.53 ± 0.14	6.27 ± 0.14	6.46 ± 0.16
Albumin (g/dL)	4.16 ± 0.11	4.30 ± 0.09	4.29 ± 0.06	4.24 ± 0.10	4.23 ± 0.11	4.25 ± 0.08
Total bilirubin (mg/dL)	0.35 ± 0.06	0.31 ± 0.07	0.39 ± 0.06	0.45 ± 0.14	0.53 ± 0.12	0.52 ± 0.09
SGOT (U/L)	101.60 ± 5.95	104.90 ± 6.20	94.20 ± 6.72	84.14 ± 7.80 ^a,b^	99.14 ± 5.98	105.90 ± 7.10
SGPT (U/L)	44.44 ± 2.56	44.10 ± 1.47	41.10 ± 2.02	43.00 ± 1.88	40.29 ± 1.76	38.90 ± 2.10
Alkaline phosphatase (U/L)	66.63 ± 5.68	70.80 ± 7.63	80.88 ± 4.15	68.50 ± 4.19	39.71 ± 5.32 ^a^	74.28 ± 10.02

Abbreviations: MCV: mean corpuscular volume; MCH: mean corpuscular hemoglobin; MCHC: mean corpuscular hemoglobin concentration; BUN: blood urea nitrogen; SGOT: serum glutamic oxaloacetic transaminase; SGPT: serum glutamic pyruvic transaminase; ALP: alkaline phosphatase. Values represent the mean ± SEM, where ^a^ indicates *p* < 0.05, which was significantly different compared to the control group; ^b^ indicates *p* < 0.05, which was significantly different compared to the vehicle group using one-way ANOVA analysis.

**Table 2 foods-12-02053-t002:** Mean body weight, the organ weight to body weight ratio, and hematologic and clinical chemistry values of the treatment study on the induced mammary tumorigenesis in rats.

Parameters	Control	Vehicle	Doxorubicin	PFPE100	PFPE200	Dox+ PFPE100	Dox+ PFPE200	Dox+ PFPE100+ Tumeric25
**Body weight** (**g**)	283.50 ± 8.71	255.33 ± 3.20	251.40 ± 5.82	277.67 ± 7.52	258.83 ± 3.79	249.33 ± 9.05	265.50 ± 9.38	249.00 ± 14.13
**Organ weight/body weight ratio**								
Heart	0.33 ± 0.01	0.33 ± 0.01	0.33 ± 0.02	0.31 ± 0.01	0.33 ± 0.01	0.29 ± 0.01	0.31 ± 0.01	0.32 ± 0.02
Kidney	0.60 ± 0.02	0.64 ± 0.01	0.67 ± 0.02	0.62 ± 0.01	0.64 ± 0.01	0.69 ± 0.01	0.69 ± 0.03	0.67 ± 0.04
Liver	2.80 ± 0.12	3.07 ± 0.09	3.69 ± 0.11 ^a^	3.21 ± 0.07	3.28 ± 0.05	4.27 ± 0.16 ^a^	4.16 ± 0.09 ^a^	3.55 ± 0.10
Lung	0.37 ± 0.01	0.41 ± 0.01	0.39 ± 0.01	0.38 ± 0.02	0.45 ± 0.04	0.40 ± 0.02	0.39 ± 0.01	0.38 ± 0.02
Spleen	0.27 ± 0.01	0.29 ± 0.02	0.17 ± 0.05 ^a^	0.27 ± 0.01	0.25 ± 0.00	0.27 ± 0.05	0.20 ± 0.04	0.12 ± 0.02 ^a^
Stomach	0.46 ± 0.01	0.50 ± 0.01	0.42 ± 0.03	0.46 ± 0.01	0.52 ± 0.02	0.52 ± 0.03	0.55 ± 0.05	0.45 ± 0.01
**Hematologic values**								
White blood cells (×10^3^/µL)	3.72 ± 0.16	4.40 ± 0.54	1.23 ± 0.41 ^a,b^	3.33 ± 0.16 ^b,c^	3.22 ± 0.14 ^b,c^	2.13 ± 1.05 ^a,b^	1.50 ± 1.53 ^a,b^	1.08 ± 0.37 ^a,b^
Neutrophil (%)	28.00 ± 2.79	42.83 ± 4.85	50.40 ± 9.53 ^a^	45.20 ± 8.65	36.17 ± 8.57	55.17 ± 6.28 ^a^	60.67 ± 7.60 ^a^	51.33 ± 4.32 ^a^
Lymphocyte (%)	71.83 ± 2.56	59.00 ± 5.84	49.20 ± 9.71	47.17 ± 8.14	60.67 ± 8.89	37.83 ± 6.55 ^a,b^	43.60 ± 6.82	47.33 ± 4.32
Monocyte (%)	0.17 ± 0.18	0.83 ± 0.29	0.00 ± 0.00	0.83 ± 0.41	0.33 ± 0.37	0.80 ± 0.37	0.67 ± 0.42	1.33 ± 0.71
Eosinophil (%)	0.83 ± 0.40	0.67 ± 0.42	0.00 ± 0.00	1.50 ± 0.81	2.17 ± 0.48 ^a,b,c^	0.00 ± 0.00	0.00 ± 0.00	0.25 ± 0.25
Basophil (%)	0.00 ± 0.00	0.00 ± 0.00	0.00 ± 0.00	0.00 ± 0.00	0.00 ± 0.00	0.00 ± 0.00	0.00 ± 0.00	0.00 ± 0.00
Red blood cell (×10^6^/µL)	7.13 ± 0.35	7.92 ± 0.32	5.17 ± 1.13 ^a,b^	7.40 ± 0.25 ^c^	7.41 ± 0.35 ^c^	5.53 ± 1.13 ^b^	5.55 ± 0.20 ^a,b^	4.62 ± 1.13 ^a,b^
Hemoglobin (g/dL)	13.82 ± 0.55	14.38 ± 0.43	9.72 ± 2.00 ^a,b^	13.97 ± 0.19	13.97 ± 0.63	10.47 ± 2.21 ^a,b^	10.68 ± 0.41 ^a,b^	9.62 ± 1.37 ^a,b^
Hematocrit (%)	36.50 ± 1.71	40.17 ± 1.48	26.00 ± 3.11 ^b^	37.00 ± 1.29	37.17 ± 1.70	31.33 ± 5.02	27.00 ± 1.24 ^b^	26.67 ± 3.76 ^b^
MCV (fL)	51.17 ± 0.48	50.83 ± 0.31	49.00 ± 0.58 ^a,b^	50.33 ± 0.42	50.17 ± 0.31	50.33 ± 1.20	50.25 ± 0.63	49.00 ± 1.08
MCH (pg)	19.00 ± 0.37	17.83 ± 0.31 ^a^	18.67 ± 0.33	18.50 ± 0.56	18.33 ± 0.33	18.33 ± 0.33	18.75 ± 0.25	18.75 ± 0.63
MCHC (g/dL)	37.50 ± 0.76	35.00 ± 0.58 ^a^	37.67 ± 0.33 ^b^	36.83 ± 1.08	36.67 ± 0.42	37.00 ± 0.58	37.75 ± 0.75	38.00 ± 1.08
Platelet (×10^3^/µL)	689.33 ± 44.85	497.67 ± 73.36	70.66 ± 512.47 ^a,b^	515.00 ± 67.45 ^c^	640.50 ± 51.83 ^c^	216.50 ± 95.50 ^a,b^	298.75 ± 46.69 ^a,b^	118.33 ± 38.40 ^a,b^
**Clinical blood chemistry values**								
BUN (mg/dL)	22.83 ± 1.25	21.17 ± 1.38	34.00 ± 7.64 ^b^	20.67 ± 1.48 ^c^	19.83 ± 0.87 ^c^	17.67 ± 3.28 ^c^	15.75 ± 1.18 ^c^	20.00 ± 5.00
Creatinine (mg/dL)	0.66 ± 0.10	0.63 ± 0.04	0.51 ± 0.16	0.68 ± 0.06	0.67 ± 0.04	0.58 ± 0.05	0.49 ± 0.04	0.54 ± 0.06
Total protein (g/dL)	6.18 ± 0.19	6.05 ± 0.20	4.37 ± 0.43 ^a^	6.00 ± 0.14 ^c^	6.03 ± 0.15 ^c^	4.70 ± 0.35 ^a^	4.55 ± 0.17 ^a^	4.88 ± 0.21 ^a^
Albumin (g/dL)	3.52 ± 0.10	3.63 ± 0.09	2.73 ± 0.37 ^a^	3.67 ± 0.08	3.73 ± 0.07	2.40 ± 0.21 ^a^	2.80 ± 0.12 ^a^	2.95 ± 0.12 ^a^
Total bilirubin (mg/dL)	0.30 ± 0.07	0.30 ± 0.04	0.42 ± 0.07	0.26 ± 0.04	0.48 ± 0.04 ^a,b^	0.50 ± 0.07	0.38 ± 0.07	0.42 ± 0.06
SGOT (U/L)	124.00 ± 15.14	149.33 ± 9.50	147.67 ± 38.05	161.00 ± 23.32	148.83 ± 10.07	109.00 ± 47.90	119.50 ± 27.64	124.75 ± 15.90
SGPT (U/L)	51.83 ± 5.07	50.33 ± 2.62	34.00 ± 10.41 ^a^	58.67 ± 6.16	50.17 ± 3.24	36.00 ± 6.00	42.00 ± 5.12	37.75 ± 7.26
ALP (U/L)	117.50 ± 9.13	216.50 ± 29.18 ^a^	30.67 ± 13.97 ^a,b^	164.00 ± 11.81	112.20 ± 13.69 ^b^	114.50 ± 19.78 ^b^	59.53 ± 12.86 ^b^	66.00 ± 9.54 ^b^

Abbreviations: MCV: mean corpuscular volume; MCH: mean corpuscular hemoglobin; MCHC: mean corpuscular hemoglobin concentration; BUN: blood urea nitrogen; SGOT: serum glutamic oxaloacetic transaminase; SGPT: serum glutamic pyruvic transaminase; ALP: alkaline phosphatase. Values represented the mean ± SEM. ^a^
*p* < 0.05 was significantly different compared to the control group; ^b^
*p* < 0.05 was significantly different compared to the vehicle group; ^c^
*p* < 0.05 was significantly different compared to the vehicle group using one-way ANOVA analysis.

**Table 3 foods-12-02053-t003:** Hematologic and clinical chemistry values of the chronic toxicity study.

Parameters	Control	Vehicle	PFPE43	PFPE86	PFPE108	PFPE43 + Tumeric11
**Body weight** (**g**)	313.30 ± 6.05	312.80 ± 6.35	314.44 ± 4.18	313.12 ± 4.18	309.71 ± 2.70	316.35 ± 5.03
**Organ weight/body weight ratio**						
Heart	0.30 ± 0.01	0.32 ± 0.01	0.31 ± 0.00	0.32 ± 0.01	0.35 ± 0.01	0.32 ± 0.00
Kidney	0.60 ± 0.01	0.59 ± 0.01	0.59 ± 0.01	0.60 ± 0.02	0.59 ± 0.01	0.61 ± 0.01
Liver	2.86 ± 0.07	2.83 ± 0.07	2.90 ± 0.04	3.11 ± 0.03	3.17 ± 0.05	2.96 ± 0.04
Lung	0.42 ± 0.02	0.42 ± 0.03	0.38 ± 0.00	0.39 ± 0.01	0.44 ± 0.01	0.41 ± 0.02
Spleen	0.24 ± 0.01	0.24 ± 0.01	0.25 ± 0.00	0.26 ± 0.01	0.24 ± 0.01	0.25 ± 0.00
Stomach	0.49 ± 0.01	0.47 ± 0.01	0.49 ± 0.01	0.52 ± 0.01	0.54 ± 0.01	0.49 ± 0.01
**Hematologic values**						
White blood cells (×10^3^/µL)	4.24 ± 0.16	3.34 ± 0.40	2.79 ± 0.11 ^a^	2.95 ± 0.19 ^a^	2.81 ± 0.20 ^a^	3.36 ± 0.26
Neutrophil (%)	19.90 ± 1.57	27.60 ± 2.83	28.35 ± 3.14 ^a^	29.65 ± 1.79 ^a^	31.50 ± 3.36 ^a^	25.58 ± 1.23
Lymphocyte (%)	72.40 ± 2.11	64.10 ± 2.89	68.59 ± 3.04	62.00 ± 1.78	64.79 ± 3.31	67.79 ± 1.69
Monocyte (%)	6.80 ± 0.79	6.70 ± 0.66	1.29 ± 0.20 ^a,b^	5.59 ± 0.52	1.79 ± 0.21 ^a,b^	5.21 ± 0.77
Eosinophil (%)	1.90 ± 0.72	1.60 ± 0.41	1.76 ± 0.13	3.35 ± 0.52 ^a,b^	2.00 ± 0.18	1.95 ± 0.28
Basophil (%)	0.00 ± 0.00	0.00 ± 0.00	0.00 ± 0.00	0.00 ± 0.00	0.00 ± 0.00	0.00 ± 0.00
Red blood cell (×10^6^/µL)	7.79 ± 0.22	7.07 ± 0.50	7.84 ± 0.09	6.69 ± 0.20	6.97 ± 0.25	7.10 ± 0.27
Hemoglobin (g/dL)	14.34 ± 0.45	13.49 ± 0.63	14.73 ± 0.15	13.04 ± 0.39	13.60 ± 0.44	13.58 ± 0.48
Hematocrit (%)	40.70 ± 1.11	37.90 ± 2.06	40.06 ± 0.57	35.88 ± 1.07	36.86 ± 1.20	37.21 ± 1.45
MCV (fL)	52.50 ± 0.39	54.20 ± 1.74	51.18 ± 0.22	53.76 ± 0.24	53.43 ± 0.47	52.32 ± 0.15
MCH (pg)	17.90 ± 0.17	18.80 ± 0.69	18.35 ± 0.11	19.06 ± 0.16	19.21 ± 0.19	18.74 ± 0.15
MCHC (g/dL)	34.20 ± 0.48	34.70 ± 0.50	35.94 ± 0.22	35.35 ± 0.23	35.79 ± 0.32	35.37 ± 0.50
Platelet (×10^3^/µL)	781.60 ± 40.07	635.30 ± 40.19	721.18 ± 21.74	647.12 ± 24.82	791.07 ± 43.22	671.47 ± 52.17
**Clinical blood chemistry values**						
BUN (mg/dL)	21.00 ± 0.73	21.10 ± 0.91	22.35 ± 0.24	23.41 ± 0.52	26.14 ± 0.73	24.11 ± 0.59
Creatinine (mg/dL)	0.61 ± 0.03	0.60 ± 0.02	0.67 ± 0.02	0.73 ± 0.02	0.54 ± 0.03	0.47 ± 0.02 ^a,b^
Total protein (g/dL)	6.59 ± 0.11	6.49 ± 0.11	6.54 ± 0.07	6.28 ± 0.07	6.42 ± 0.07	6.49 ± 0.06
Albumin (g/dL)	3.72 ± 0.07	3.87 ± 0.06	3.74 ± 0.03	3.78 ± 0.03	3.69 ± 0.03	3.81 ± 0.03
Total bilirubin (mg/dL)	0.33 ± 0.06	0.26 ± 0.02	0.30 ± 0.05	0.26 ± 0.04	0.14 ± 0.05	0.25 ± 0.03
SGOT (U/L)	157.80 ± 5.83	142.40 ± 8.13	165.65 ± 2.80	140.59 ± 8.91	108.21 ± 3.82 ^a,b^	118.32 ± 6.44
SGPT (U/L)	63.70 ± 3.82	61.10 ± 4.30	60.06 ± 3.17	64.41 ± 4.15	66.43 ± 2.58	60.32 ± 3.63
ALP (U/L)	98.60 ± 10.88	119.10 ± 32.05	102.06 ± 3.00	105.65 ± 5.89	142.07 ± 7.22 ^a^	113.74 ± 7.11

Abbreviations: MCV: mean corpuscular volume; MCH: mean corpuscular hemoglobin; MCHC: mean corpuscular hemoglobin concentration; BUN: blood urea nitrogen; SGOT: serum glutamic oxaloacetic transaminase; SGPT: serum glutamic pyruvic transaminase; ALP: alkaline phosphatase. Values represent the mean ± SEM. ^a^
*p* < 0.05 was significantly different compared to the control group; ^b^
*p* < 0.05 was significantly different compared to the vehicle group using one-way ANOVA analysis.

## Data Availability

All data represented in this study are available upon request from the corresponding authors.

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
