# Peer review of "Piper nigrum Extract: Dietary Supplement for Reducing Mammary Tumor Incidence and Chemotherapy-Induced Toxicity"

_foods, 2023, doi:10.3390/foods12102053_

Round 1

Reviewer 1 Report (Previous Reviewer 1)

This revised manuscript has been significantly improved based on the suggestions of the reviewer. While there may be some information or parts that are not yet fully satisfactory. However, it is essential to have the manuscript proofread by an English language expert before the final publication.

This revised manuscript has been significantly improved based on the suggestions of the reviewer. While there may be some information or parts that are not yet fully satisfactory, it is suitable for publication in its current format. However, it is essential to have the manuscript proofread by an English language expert before the final publication.

Author Response

Reviewer 2 Report (New Reviewer)

The several points must be improved in this manuscript:

1. Please add a brief sentence to make clearer regarding 'support the body during breast cancer treatment' (lines 16-17). 

2. Please write alphabetically of the keywords.

3. Please write in consistently to write Piper nigrum L (in italic form) in whole manuscript. Some words have been written in italic (line 100).

4. Why did the author not use the same animal in these studies? 

5. The critical point of this manuscript is the treatment of test group was the PFPE-CH as individual without combining with vehicle group? I think in the application they should combine with vehicle group, as the author mentioned the reason using 5% cold-pressed coconut oil in this study (lines 511-517). Please make sure about this manner.

6. Please add new data, correlation bioactive compounds that showed in Table S1 with in-vivo data experiment.

7. Please relate the expression in the introduction (lines 47-51) regarding three types of cancer prevention. Please consider, what type of prevention were found in this study?

Author Response

Reviewer 3 Report (New Reviewer)

The manuscript deals with the evaluation of Piper nigrum extract as potential anticancer agent in breast tumor therapy. The paper is well organized. The research question is well-defined, and the study addresses a knowledge gap in the field. Additionally, the multifactorial approach used in the study is appropriate for assessing the role of Piper nigrum extract in breast tumor reduction in rats as a model system. This study has a practical potential to conduct more in-depth analysis in human carcinogenesis. Therefore, I have no substantial comments related to the quality of the manuscript. Minor comments are listed below:

L65: Piper nigrum in italics throughout the paper

L181: control instead normal

L283: in Fig. 2A, vertical lines of SEM should be added

L514-516: Extend the sentence related to the health-promoting properties of honey. Include composition of vitamins, phenolic acids, flavonoids, proteins, carbohydrates, amino acids, royal jelly aliphatic acids. For this purpose the Authors should refer to the following references: https://doi.org/10.2478/jas-2018-0012, https://doi.org/10.3390/ijerph20032458

Author Response

This manuscript is a resubmission of an earlier submission. The following is a list of the peer review reports and author responses from that submission.

Round 1

Reviewer 1 Report

The authors studied about the chemo-preventive effects and safety of PFPE in rodents. The animal studies were followed by OECD guidelines and the data presentation and conclusions were appropriate. This study contains a lot of data which are mostly convincing. However, some data and writing needs to be upgraded. There are some questions and comments

The vehicle group showed tumor volume reduction in Fig. 5.

In Fig 6, indicate using arrow for ductal carcinoma or invasive ductal carcinoma. Is there 40X magnification for all figures?

Fig. 3, 4 and 8: in the legends or groups, denotes treatment groups I to VI.

Fig 4, 7, 8: Is there any statistical significance?

Cytokines and TBARS are not exactly matching the animal tumor incidence data.

It is better to show tumor tissues and PFPE-CH treated breast tissues as supplement data.

Please clearly write the limitation of this study for future use of PFPE-CH as a cancer prevention supplement in the discussion and conclusion sections.

There is hyphen in one word such as line 39 ‘prin-cipal’. Check throughout the manuscript.

Reviewer 2 Report

Dear Authors

Some general compounds

Kindly provide brief materials for eg. sub-section 2.1. Preparation of PFPE and formula

Kindly mention plant identification.

Since the authors have clearly mentioned in the introduction the phytoconstituents of the plant used in the study. What is the use of analyzing its Phytochemicals by GC-MS? I feel this repetition.

Kindly remove Table 1 from the main text and give it as supplementary files as the GC-MS results are well-known facts of the P. nigrum.

Please Don’t use the term low piperine fractional Piper nigrum; I suggest the authors change the title accordingly.

Though the present studies technically sound good, it completely lacks interest scientifically. I suggest the authors to isolate minor compounds viz., piperic acid, piperlonguminine, pellitorine, piperolein B, piperamide, piperettine, and (-)-kusunokinin in the extract and study their efficacy in animal models which will be more meaning for the future drug discovery programs instead of repeating the same experiments.